# Targeting the Tumor Microenvironment through mTOR Inhibition and Chemotherapy as Induction Therapy for Locally Advanced Head and Neck Squamous Cell Carcinoma: The CAPRA Study

**DOI:** 10.3390/cancers14184509

**Published:** 2022-09-17

**Authors:** Diane Evrard, Clément Dumont, Michel Gatineau, Jean-Pierre Delord, Jérôme Fayette, Chantal Dreyer, Annemilaï Tijeras-Raballand, Armand de Gramont, Jean-François Delattre, Muriel Granier, Nasredine Aissat, Marie-Line Garcia-Larnicol, Khemaies Slimane, Benoist Chibaudel, Eric Raymond, Christophe Le Tourneau, Sandrine Faivre

**Affiliations:** 1Department of Otorhinolaryngology, Bichat University Hospital, Université Paris Cité, 75018 Paris, France; 2Medical Oncology Department, Saint-Louis Hospital, Université Paris Cité, 75010 Paris, France; 3Medical Oncology Department, Paris-St Joseph Hospital, 75014 Paris, France; 4Institut Claudius Regaud, 31000 Toulouse, France; 5Centre Léon Bérard, 69000 Lyon, France; 6GERCOR, 75011 Paris, France; 7AFR Oncology, 92012 Boulogne-Billancourt, France; 8Novartis Pharma SAS, 92063 Rueil-Malmaison, France; 9Department of Drug Development and Innovation (D3i), Institut Curie, INSERM U909 Research Unit, Paris-Saclay University, 75005 Paris, France

**Keywords:** tumor microenvironment, mTOR inhibition, head and neck carcinoma, PI3K-AKT-mTOR pathway, interferon gamma

## Abstract

**Simple Summary:**

The PI3K-AKT-mTOR pathway is dysregulated in 70% of head and neck squamous cell carcinoma (HNSCC) and linked to the tumor microenvironment. This weekly induction treatment combined the mTOR inhibitor everolimus with carboplatin-paclitaxel chemotherapy for locally advanced T3-4/N0-3 HNSCC. In 41 patients, safety profile was favorable and overall response rate was 75.6%. Translational data demonstrated specific target engagement with p-S6K decrease in tumor tissue and pro-immunogenic cytokine release in peripheral blood. Induction treatment with chemotherapy and mTOR inhibitors may provide new therapeutic options and rationale for combinations with immune oncology agents for locally advanced HNSCC.

**Abstract:**

Mammalian target of rapamycin (mTOR) regulates cellular functions by integrating intracellular signals and signals from the tumor microenvironment (TME). The PI3K-AKT-mTOR pathway is activated in 70% of head and neck squamous cell carcinoma (HNSCC) and associated with poor prognosis. This phase I-II study investigated the effect of mTOR inhibition using weekly everolimus (30 mg for dose level 1, 50 mg for dose level 2) combined with weekly induction chemotherapy (AUC2 carboplatin and 60 mg/m^2^ paclitaxel) in treatment-naïve patients with locally advanced T3-4/N0-3 HNSCC. Patients received 9 weekly cycles before chemoradiotherapy. Objectives were safety and antitumor activity along with tissue and blood molecular biomarkers. A total of 50 patients were enrolled. Among 41 evaluable patients treated at the recommended dose of 50 mg everolimus weekly, tolerance was good and overall response rate was 75.6%, including 20 major responses (≥50% reduction in tumor size). A significant decrease in expression of p-S6K (*p*-value: 0.007) and Ki67 (*p*-value: 0.01) was observed in post-treatment tumor tissue. Pro-immunogenic cytokine release (Th1 cytokines IFN-γ, IL-2, and TNF-β) was observed in the peripheral blood. The combination of everolimus and chemotherapy in HNSCC was safe and achieved major tumor responses. This strategy favorably impacts the TME and might be combined with immunotherapeutic agents.

## 1. Introduction

Head and neck squamous cell carcinoma (HNSCC) is the most frequent tumor of the head and neck region and the sixth most frequent cancer worldwide. The main risk factors are tobacco/alcohol consumption and human papillomavirus (HPV) infection [1]. Locally advanced HNSCC is a candidate for concomitant chemoradiotherapy with platinum-based chemotherapy or cetuximab.

Induction chemotherapy is still a matter of debate [2]; however, this approach could contribute to multidisciplinary strategies against tumor recurrence. The overall response rate (ORR) using induction chemotherapy with cisplatin, 5FU, and docetaxel (TPF) in phase III trials is 68–72% [3,4].

Seeking induction regimens with more manageable safety and weekly administration of carboplatin and paclitaxel as induction chemotherapy was investigated across phase II trials. In a study by Vokes et al., 69 patients received six weekly doses of carboplatin (AUC2) and paclitaxel (135 mg/m^2^), yielding an 87% objective response rate; the most common grade 3 or 4 toxicity was neutropenia (36%) [5]. Another trial by Ready et al. using the same dose and schedule in 35 patients reported an overall response rate with induction of 79% [6]. The more manageable safety profile of this weekly combination makes it an ideal backbone for candidate combination regimens.

Mammalian target of rapamycin (mTOR) is a serine/threonine kinase acting downstream of the activation of phosphatidylinositol 3-kinase (PI3K) [7]. Pathological activation of the PI3K-AKT-mTOR pathway has been reported in >70% of HNSCC and is associated with poor prognosis [8]. It is a mechanism of resistance to platinum compounds, and the addition of mTOR inhibitors to platinum-based chemotherapy could potentiate the proapoptotic effects of platinum compounds and taxanes in HNSCC cells [8,9,10,11]. Moreover, in the tumor microenvironment (TME), the activity of most immune cell types is affected by the PI3K-AKT-mTOR pathway [12,13]. Indeed, mTOR signaling interacts with TME through programmed cancer cell death axis (PD-L1/PD1) that is stimulated by PTEN deletion or loss of function [14,15], as well as effects on T cell differentiation [16,17]. Thus, besides its tumor-targeted mode of action, mTOR inhibition may also affect the antitumor immune response within the TME.

The CAPRA study (CArboplatin, Paclitaxel, RAD001) was a phase I-II study designed to assess the feasibility of adding everolimus (previously denominated RAD001) to weekly carboplatin and paclitaxel as induction chemotherapy in patients with untreated locally advanced HNSCC. We used a nine-weekly administration schedule to match the 9 weeks duration of usual chemotherapy induction with three cycles of TPF. CAPRA was followed by concurrent chemoradiotherapy whenever possible. The maximum tolerated and recommended phase II dose of everolimus when administered in combination with carboplatin and paclitaxel was evaluated. Analyses of TME-derived blood and tumor tissue biomarkers were performed to identify patients who may derive most benefit from this combination.

## 2. Materials and Methods

### 2.1. Eligibility Criteria

All patients had histologically proven squamous cell carcinoma of the oral cavity, oropharynx, larynx, or hypopharynx. Tumors were unresectable locally advanced disease (T3-4/N0-N3), or resectable but with contra-indication to surgery. Eligible patients had not received prior chemotherapy or radiotherapy; had a WHO performance status (PS) of 0 to 2; were older than 18 years; had no uncontrolled infection; and had adequate hematologic (neutrophil count ≥ 1500/mL; platelet count ≥ 100,000/mL), renal (serum creatinine ≤ 3 mg/dL or clearance ≥ 40 mL/min), and hepatic (bilirubin ≤ 1.5 × the upper limit of normal values and alkaline phosphatases ≤ 5 × the upper limit of normal) functions. Presence of distant metastases was an exclusion criterion. The study was conducted in accordance with ICH Good Clinical Practice guidelines and the Declaration of Helsinki. The protocol was approved by the institutional ethics committee at each participating center, and written informed consent was obtained from each patient. The clinical trial was registered: NCT01333085.

### 2.2. Study Design and Treatment

This study used a standard 3 + 3 dose escalation design, with weekly paclitaxel 60 mg/m^2^ over 1 h and carboplatin area under the curve 2 (AUC2) over 1 h by intravenous infusion, then escalating doses of weekly oral everolimus starting at the dose of 30 mg/week and escalating up to 50 mg/week, 1 h before or 2 hours after lunch. Treatment was given for nine consecutive weekly cycles or until clinical evidence of disease progression or unacceptable toxicity. Before paclitaxel infusion in weeks 1–3, premedication with dexchlorpheniramine 5 mg was recommended. After completion of induction chemotherapy, patients could either undergo exclusive radiotherapy; concomitant radio-chemotherapy with either cisplatin 100 mg/m^2^ on days 1, 22, and 43 or weekly cetuximab; or surgery. A minimum of 3 patients were enrolled at each dose level and followed for 4 weeks before accrual of other patients to the next higher dose level. If a patient withdrew before completing 7 days of therapy without experiencing a dose-limiting toxicity (DLT), an additional patient was added to the dose level. Patients with a treatment delay of ≥2 weeks due to adverse events (AEs) were not replaced (considered to be DLT). If no patients experienced a DLT, enrolment was performed at the next dose level. If 1 out of 3 patients experienced a DLT, the cohort was expanded to 6 patients; if no additional DLTs were observed, dose could be escalated in next cohort enrolled; if ≥1 additional DLT was observed, dose would be considered to be the maximum tolerated dose (MTD). DLT was defined during the first 4 weeks of treatment as any of the following: absolute neutrophil count <0.5 × 10^9^ persisting for ≥7 days; febrile neutropenia; grade 3 thrombocytopenia persisting for ≥7 days; platelets <25 × 10^9^/L; bleeding considered imputable to thrombocytopenia; grade ≥3 diarrhea despite optimal treatment; grade ≥3 rash (or grade 2 if medically concerning or unacceptable to the patient); grade 3 or >7 days grade 2 non-infectious pneumonitis; other grade ≥3 AEs considered imputable to treatment and persisting for ≥7 days despite optimal treatment, or recurring during the same cycle; any AE requiring a delay to the next treatment cycle of ≥2 weeks; any AE necessitating reduced doses of paclitaxel to <45 mg/m^2^ or carboplatin to <AUC1.5. Phase II expansion patients received 50 mg/week everolimus after assessment of tolerability in phase I. Cycle delay was allowed up to 14 days for each weekly administration. Dose adaptations of each drug were allowed according to the following next lower weekly dose level: carboplatin AUC1.5 instead of AUC2, paclitaxel 45 m/m^2^ instead of 60 mg/m^2^, everolimus 30 mg instead of 50 mg. The phase II expansion was conducted according to a Simon two-stage design, which required at least 16 of the initial 25 patients to display objective tumor response at 12 weeks before proceeding with enrollment to a total of 45 patients.

### 2.3. On-Study Evaluation

Safety parameters assessed at baseline and study visits included physical examination, vital signs, WHO performance status, hematology, blood chemistry, and urinalysis. Patients were examined weekly during phase I of the study, and every 3 weeks during phase II; final assessment was performed within 14 days of last treatment and completion of radiation therapy. Toxicity was monitored throughout treatment and graded according to the National Cancer Institute Common Terminology Criteria for Adverse Events (NCI-CTCAE, V3.0). Cervicothoracic CT scans were performed at screening, and at the end of induction chemotherapy according to protocol (week 11) to evaluate response. Tumor response was assessed by the investigator according to Response Evaluation Criteria in Solid Tumors (RECIST, v1.1). Subsequent locoregional therapy was reported whenever evaluable from medical reports.

### 2.4. Endpoints and Statistics

For the phase I part, the primary objective was to evaluate the safety and determine the weekly recommended dose of everolimus to be combined with weekly carboplatin and paclitaxel. For the phase II expansion part, the primary endpoint was overall response rate (ORR) according to RECIST after completion of induction treatment for all patients treated with the recommended dose of everolimus. All treated patients were included in the safety analysis. Additional secondary objectives included toxicity profile and translational research assessing tumor and blood biomarkers aiming at identifying sensitivity/resistance factors to induction treatment. With consideration to published reports, a 60% 9-week ORR was considered unsufficient (P0) and 80% ORR was considered as clinically relevant (P1). A sample size of 45 patients was calculated on the basis of the Simon two-stage design to provide 90% power with 0.05 significance level for testing. If at least 34 (75.5%) patients experienced an objective response, the combination therapy would be considered effective.

### 2.5. Translational Research

In one patient with a technically accessible tumor, an explanatory ex vivo pharmacodynamic study was performed on a pre-treatment biopsy. Fresh tumor tissue was 300 μm sliced, then incubated during 48 h with 0.1 µM everolimus or dimethylsulphoxide (DMSO; control). Ki67, caspase 3, and phosphorylation of downstream S6 kinase 1 (p-S6K) expression were evaluated by immunohistochemistry (IHC) before and after exposure.

In nine patients, biopsies were performed at screening and at the end of induction therapy (week 9) and reviewed centrally in the pathology department at Beaujon University Hospital, Clichy-La-Garenne, France. Biomarkers were evaluated by IHC or immunofluorescence (IF), using Ki67 and p-S6K.

For IHC studies realized with BenchMark (Ventana Medical Systems), samples were deparaffinized, embedded with Cell conditionning 1 (Ventana Medical Systems, 950-124, Oro Valley, United-States), and incubated with the primary and secondary antibodies. The following antibodies were used: p-S6K (Sigma Aldrich # S6311, 1:200, Darmstadt, Germany), Ki67 (Dako, #M7240; 1:200), and caspase 3 (Cell Signaling Technology # 9662, 1:400, Leiden, Netherlands).

Blood samples (5 mL) were collected before starting induction therapy (week 0) and at weeks 1, 4, and 9 during induction therapy. The following blood biomarkers were evaluated: interleukin (IL)-1alpha, IL-1beta, tumor necrosis factor (TNF)-beta, IL-8, IL-12, TNF-alpha, granulocyte-macrophage-colony-stimulating factor (GM-CSF), IL-6, IL-12, IL-2, IL-4, IL-5, IL-13, IL-15, IL-17A, vascular endothelial growth factor (VEGF), IL-17A, and interferon (IFN)-gamma. Results of biomarker assessments were binary or semi-quantitatively expressed, and correlative relationships with clinical data were explored using Fisher’s exact or chi-squared tests.

Multivariate analysis was performed with a Cox proportional hazards regression model. Variables that were associated with DFS in univariate analysis with a *p* < 0.10 and/or had a known prognostic value were included in the model. Logistic regression was performed using the Logit method. Statistical significance was defined as *p* < 0.05.

## 3. Results

### 3.1. Patient Characteristics

Fifty patients were enrolled at five centers. A total of 7 patients participated in the phase I dose-escalation study, and 43 patients participated in the phase II expansion study. Among the four patients treated at the first dose level of 30 mg/week of everolimus, one patient was overdosed due to self-misunderstanding (30 mg/day, 3 consecutive days at first cycle) and therefore was withdrawn early from the study and not evaluable for neither safety nor activity. A total of 46 patients (3 in dose escalation, 43 in expansion) were treated at the predefined maximum dose of 50 mg/week and were evaluable for safety (Figure 1).

Baseline characteristics of the three evaluable patients treated at the first dose level of 30 mg/week are summarized in Appendix A. Demographics and baseline characteristics of the 46 patients treated at the recommended dose are summarized in Table 1. Median age was 58 years, and most patients had stage IVa disease, 39 of them with active tobacco consumption > 20–30 pack-year.

### 3.2. Dosing and Toxicity

Seven patients were included in the phase I dose escalation part of the study. No DLT was reported, even for the patient who was overdosed. Everolimus 50 mg/week was the highest dose assessed in combination with carboplatin and paclitaxel, therefore identified as the recommended for the phase II cohort expansion. A total of 46 patients were treated at the recommended dose of everolimus of 50 mg/week. Those patients received a total number of 325 cycles with a median number of 7 cycles per patient. A total of 18 patients had a cycle delay beyond 7 days, among them, 8 had a cycle delay beyond 14 days. Dose reductions, mainly due to hematological toxicity, were necessary in 6 patients for carboplatin, 7 patients for paclitaxel, and 14 patients for everolimus.

The combination was generally well tolerated, and 83% of AEs were grade 1–2 (Table 2). Hematologic toxicities were manageable, and patients did not experience febrile neutropenia nor bleeding. The most common AEs reported with the overall treatment combination were asthenia, nausea, alopecia, and mucositis. Everolimus-associated AEs included grade 1–2 hypercholesterolemia (27 patients) and grade 3 rash (1 patient), pruritus (1 patient), dyspnea (1 patient), and hyperglycemia (2 patients). One patient died of cardiac arrest, which was deemed unrelated to study treatment; no toxic death was reported.

### 3.3. Activity

Forty-one patients were evaluable for antitumor activity. Five patients were not evaluable for activity since imaging was not performed due to the following reasons (each in one patient): patient choice to discontinue the study, intercurrent peritonitis, intercurrent thromboembolic event with concomitant biological toxicity, investigator choice to discontinue the study due to hematological toxicity, and death related to cardiac arrest mentioned above. Overall response rate was 75.6%, and thus the combination therapy was considered effective according to predefined statistical hypothesis. The triplet regimen was associated with a high antitumor activity (Table 3, Figure 2). Twenty patients experienced major responses (≥50% reduction in tumor size). Noticeably, major responses were observed in several patients with bulky necrotic and hypoxic lesions (Figure 3). Regarding the faisability of subsequent locoregional therapy, among 46 patients who received everolimus at 50 mg/week, 41 patients underwent platinum-based chemoradiotherapy after CAPRA treatment.

### 3.4. Translational Research

#### 3.4.1. Ex Vivo Pharmacodynamic Study

In one patient included in the study with accessible tumor, part of the pre-treatment tumor biopsy sample was sliced and exposed to everolimus ex vivo. Compared to the control slices, exposure to everolimus decreased the proliferation of tumor cells with decreased Ki67 expression and induced apopotosis with increased caspase 3 expression. A significant decrease in staining for p-S6K, a phosphorylated downstream target of the mTOR signaling pathway, was observed with everolimus, suggesting specific target engagment (Figure 4).

#### 3.4.2. Pharmacodynamic Histological Biomarkers

For nine patients, pre- and post-treatment biopsies were technically available for comparative analysis. A significant decrease in p-S6K staining was observed in post-treatment biopsies as compared to baseline (*p*-value: 0.007, Figure 5), suggesting specific target engagement and inhibition of mTOR signaling by everolimus. Ki67 levels were also decreased (*p*-value: 0.01), suggesting decreased tumor proliferation.

#### 3.4.3. Blood Biomarkers

Forty-three patients were evaluable for sequential analysis of cytokine levels in blood. Given the limited number of patients and the amplitude of the variations, analysis did not reach statistical significance; the variation profiles are displayed in Figure 6. Induction treatment was associated with a trend of early increase in plasma levels of pro-immunogenic Th1 cytokines IFN-γ, IL-2, and TNF-β, as well as a trend of transient decrease in IL-10, suggesting a facilitating role for antitumor immunity at early timepoints. A secondary trend of increase in suppressive cytokines IL-4, IL-5, and IL-10 was observed at later timepoints.

## 4. Discussion

Although very efficient, the toxicity associated with standard-of-care induction chemotherapy for locally advanced HNSCC may compromise the optimal administration of subsequent chemoradiotherapy, especially due to taxane-associated myelotoxicity [18]. Therefore, innovative induction regimens must improve tolerability. The chemotherapeutic agents and the weekly schedule selected in CAPRA aimed at optimizing tolerability without decreasing efficacy. As such, carboplatin was preferred to cisplatin because it has better tolerability with a similar effectiveness [19,20]. Addition of a taxane to platinum-based induction chemotherapy has been shown to improve efficacy [4,21]. Moreover, paclitaxel displays similar clinical efficacy and has more easily manageable toxicities than docetaxel and has shown synergistic effects with carboplatin and mTOR inhibitors in preclinical studies [11]. Targeting the PI3K-AKT-mTOR pathway has become an attractive option for treatment of HNSCC, with several agents being currently tested in clinical trials, both as single agents and in combination [22]. Our study demonstrates that the CAPRA combination regimen with weekly carboplatin, paclitaxel, and everolimus 50 mg is highly effective as induction therapy in patients with previously untreated, locally advanced HNSCC. Efficacy of this treatment compares aptly with that of TPF, a triplet chemotherapy based on a docetaxel, cisplatin, and 5-FU combination (75.6% ORR for CAPRA vs. 68–72% for TPF) [3,4] and to phase II results using weekly carboplatin and docetaxel (79–87% ORR) [5,6]. Clinical toxicities compared favorably to those of TPF even if CAPRA was associated with increased haematological toxicities that were easily manageable. Of note, CAPRA did not compromise further locoregional approaches since 41 patients (84%) were able to receive platinum-based chemoradiotherapy after induction with CAPRA.

In HNSCC, activation of the PI3K-AKT-mTOR pathway is the most frequently dysregulated signaling pathway [23,24,25]. The mTOR inhibitors have been shown to display anti-lymphangiogenic properties in preclinical models [26]. This antiangiogenic process relies on effects on pericytes and endothelial cells rather than on cancer cells themselves [27]. Everolimus might thus act favorably against the first route of tumor dissemination to cervical lymph nodes, which is the most significant predictor of tumor recurrence in HNSCC.

Translational research in our study showed that everolimus yielded direct and specific inhibition of the PI3K-AKT-mTOR pathway in tumor tissues with a significant decrease in p-S6K observed in a majority of post-treatment biopsies as compared to baseline biopsies. Among several surrogate markers assessing mTOR inhibition in skin, peripheral blood mononuclear cells, and/or in the tumor cells, a decrease in the p-S6K has been reported as the most reliable pharmacodynamical biomarker in preclinical studies and clinical trials [8]. As such, our study confirms the expected inhibition of mTOR pathway in HNSCC of patients treated with everolimus. Due to limited material from paired biopsies, we had to select a limited number of biomarkers. Unfortunately, tumor tissue available did not allow for the inclusion of p-AKT as an additional parameter of interest to assess the compensatory feedback loop through the mTORC2-AKT pathway, following mTORC1 inhibition with everolimus [8].

HNSCCs deploy multiple mechanisms to avoid immune recognition and subsequent antitumor immune response. The recruitment of mononuclear-myeloid-derived suppressor cells (MDSCs) and conditioning of the surrounding TME by expressing immune suppressive chemokines and cytokines, leading to the accumulation of suppressive regulatory T cells (Tregs) and the polarization of tumor associated macrophages toward an immune suppressive M2 phenotype, are the main processes [28]. At the same time, rapalogs (rapamycin and its analogs) have been used as a long-term immunosuppressive treatment in solid organ transplant because of their properties to counteract T cell activation [8]. However, mTOR inhibitors may have both pro- and anti-inflammatory actions and prompt both initial stimulation of Type 1 T helper (Th1) CD4+ T cells as well as long-term expansion of Tregs in cancer [29,30]. Unsurprisingly, cytokine plasma level analyses in patients treated with CAPRA were complex, with significant timely variations. Time may be of the essence. A most prominent feature in our study was an early but transient increase in Th1 cytokine expression, particularly IFN-γ, as well as a transient decrease in IL-10. This suggests that the initial effect of everolimus could be to promote a Th1-cytokine-enriched TME before its immunosuppressive properties overcome this early benefit, as illustrated in our study by later increases in IL-4, IL-5, or IL-10. These modifications may have contributed to initial treatment efficacy by enhancing local immunity. Of note, a Th1-cytokine-enriched TME and high levels of IFN-γ are biomarkers for anti-PD-1 efficacy in many cancer types including HNSCC [31,32,33]. Thus, association of everolimus-containing regimen such as CAPRA with an anti-PD-1 therapy should be evaluated preferentially in contexts where a short-course treatment is needed, such as induction therapy for unresectable disease [25,34].

Several questions remain open given the limitations of our study. First, there was no clear, established advantage of everolimus-carboplatin-paclitaxel treatment over carboplatin-paclitaxel or TPF treatment ORR, and the respective impact of chemotherapy and everolimus could not be assessed in this single-arm study. Moreover, our study did not plan systematic HPV status testing. One limitation of our trial is thus the lack of a significant subgroup of patients with HPV-related HNSCC without other risk factors. Indeed, alterations of the PI3K pathway appear particularly prevalent in HPV+ tumors [23,24]. These tumors, which have an increased risk of distant failure when relapsing, may be particular candidates to clinically and biologically optimized therapy regimens such as the CAPRA combination, which deserves to be further tested in this subpopulation. Due to limited material from paired biopsies, we had to select a limited number of biomarkers. Unfortunately, the tumor tissue available did not allow for us to include p-AKT as an additional parameter of interest to assess the compensatory feedback loop through the mTORC2-AKT pathway, following mTORC1 inhibition with everolimus [8].

Data obtained from translational research in this trial allowed us to report descriptive variations of tumor and blood biomarkers rather than identifying sensitivity/resistance factors to induction treatment. In addition, we could not compare our findings with biomarker data in patients receiving only carboplatin-paclitaxel chemotherapy, with those data not being available in previously published studies. Whether the weekly everolimus regimen may favor a Th1-shifted balance more than a daily regimen is another interesting question.

## 5. Conclusions

Weekly everolimus with carboplatin/paclitaxel as an induction regimen was well tolerated and yielded a high rate of objective responses in patients with locally advanced HNSCC. Translational data showed the target engagement of the mTOR pathway that was associated with changes in TME and proliferation of tumor cells. Since there was no clear advantage of CAPRA over carboplatin-paclitaxel or TPF treatments in terms of ORR, future directions may investigate the proper role of everolimus toward potential lower recurrence rate, or better response in a subset of tumors showing hyperactivation of the PI3K-AKT pathway.

## Figures and Tables

**Figure 1 cancers-14-04509-f001:**
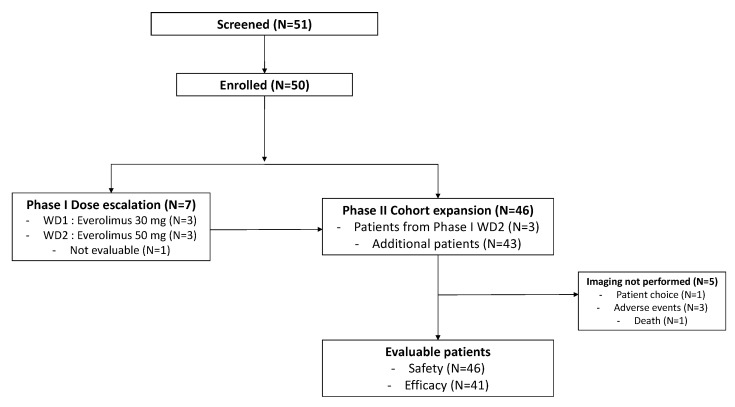
Flow chart (WD: weekly dose).

**Figure 2 cancers-14-04509-f002:**
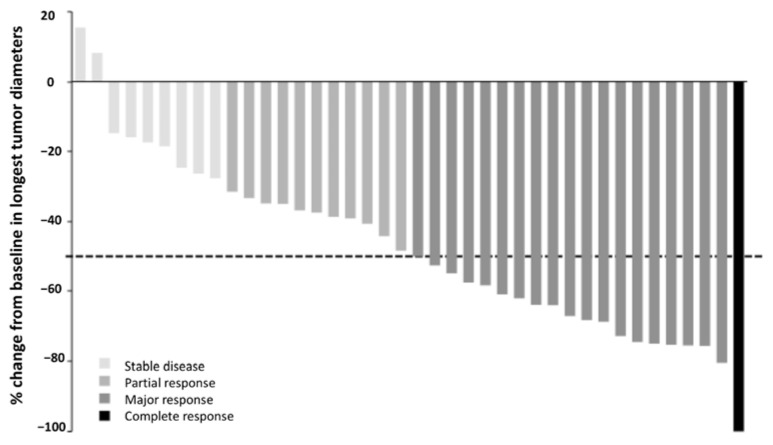
Waterfall plot of radiological response per RECIST v1.1 in non-progressive patients (N = 40). The dotted line depicts the 50% cut-off for major response.

**Figure 3 cancers-14-04509-f003:**
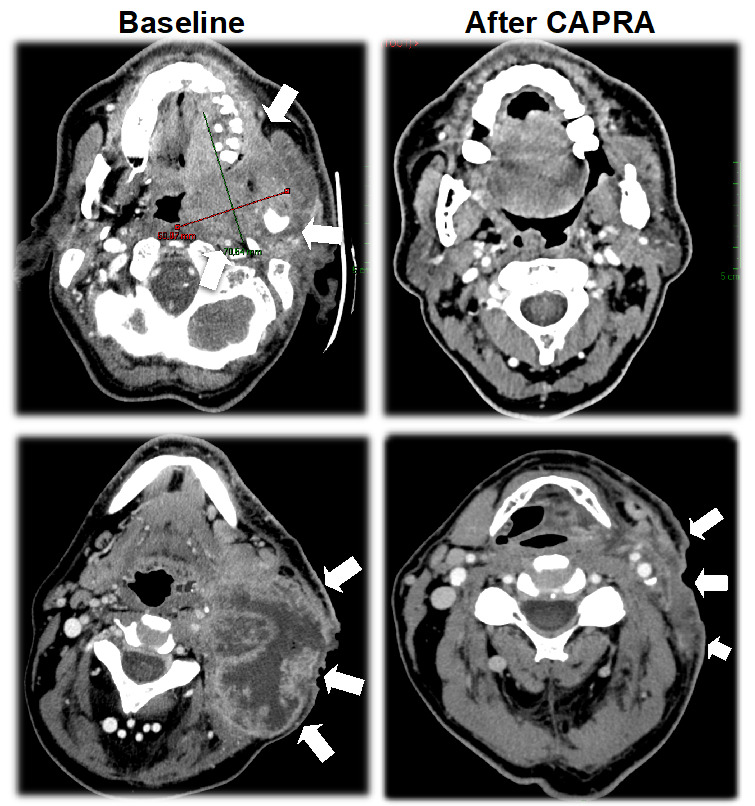
Imaging evolution at baseline and after 9 cycles of induction treatment showing major response in a case of oropharyngeal squamous cell carcinoma (upper panels) and another case of N3 necrotic cervical node involvement of hypopharyngeal squamous cell carcinoma (lower panels). The white arrows show the tumor localization on each slide.

**Figure 4 cancers-14-04509-f004:**
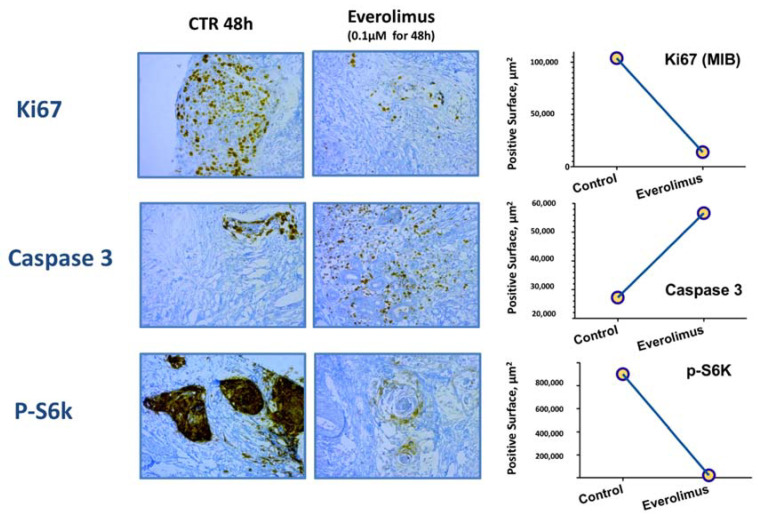
Ex vivo study on fresh pre-treatment tumor tissue: Ki67, caspase 3, and p-S6K immunohistochemistry before and after 48 hours exposure with 0.1 µM of everolimus or with dimethylsulphoxide for controls (CTR). Y-axis scale showing positive staining surface in μm^2^ versus total histological slide surface.

**Figure 5 cancers-14-04509-f005:**
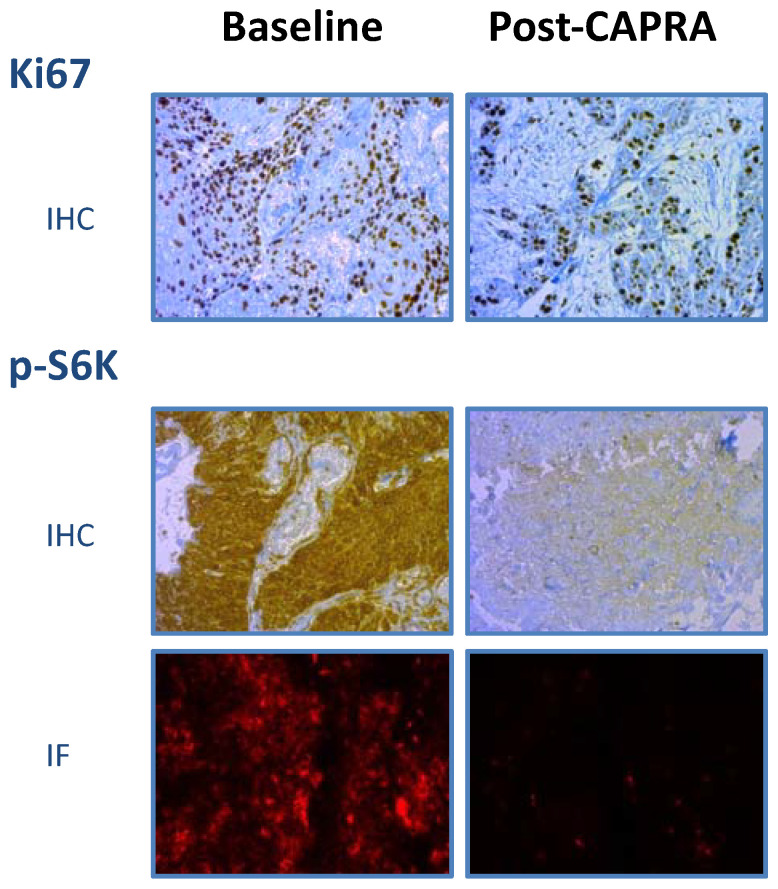
Pharmacodynamic histological biomarkers: representative staining of Ki67 immunohistochemistry and p-S6K immunohistochemistry and immunofluorescence at baseline and after 9 cycles of induction treatment (N = 9). IF: immunofluorescence, IHC: immunohistochemistry.

**Figure 6 cancers-14-04509-f006:**
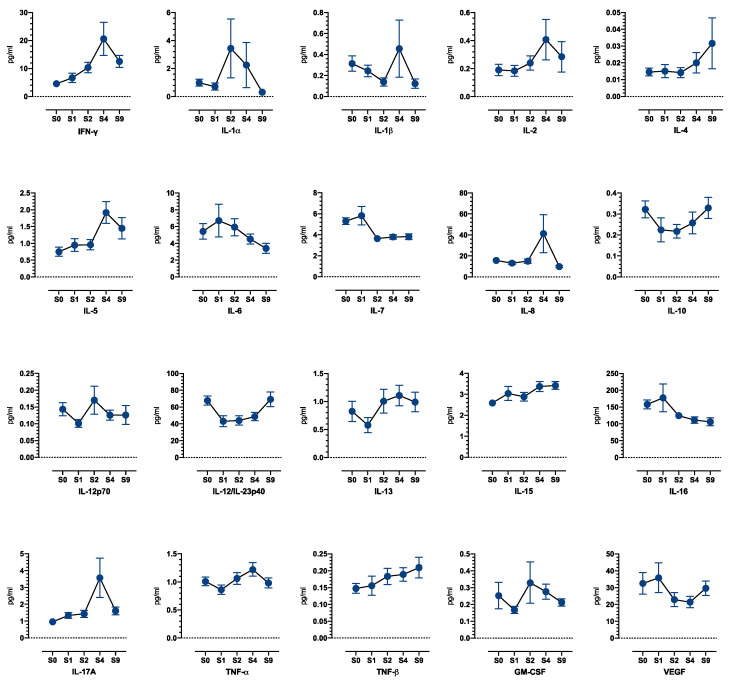
Blood biomarkers: Cytokines protein expression profile at baseline and after 9 cycles of induction treatment (N = 43).

**Table 1 cancers-14-04509-t001:** Patient demographics and baseline characteristics of patients treated at the recommended dose of everolimus of 50 mg/week.

Characteristics	
Number of patients enrolled	N = 46
Age, median (year); (range)	58 (39–85)
Gender, N (%)	
Male	37 (80.4)
Female	9 (19.6)
WHO performance status, n (%)	
0	22 (47.8)
1	18 (39.1)
2	6 (13.0)
Disease stage, N (%)	
IVa	30 (65.2)
IVb	16 (34.8)
T stage, N (%)	
1	2 (4.3)
2	6 (13.0)
3	2 (4.3)
4a	31 (67.4)
4b	4 (8.7)
x	1 (2.2)
N stage, N (%)	
1	6 (13.0)
2a	3 (6.5)
2b	5 (10.9)
2c	14 (30.4)
3	13 (28.3)
x	5 (6.5)
Primary site, N (%)	
Oropharynx	24 (52.2)
Oral cavity	13 (28.3)
Hypopharynx	5 (6.5)
Larynx	4 (8.7)

**Table 2 cancers-14-04509-t002:** Treatment-emergent adverse events observed in patients treated at the recommended everolimus dose (N = 46).

Adverse Event	All Grades, N (%)	Grade 1–2, N (%)	Grade 3–4, N (%)
**Hematologic toxicity ^1^**			
Leucopenia	39 (85)	26 (57)	13 (28)
Neutropenia	40 (87)	16 (35)	24 (52)
Anemia	43 (93)	35 (76)	8 (17)
Thrombocytopenia	37 (80)	31 (67)	6 (13)
**Biologic toxicity ^2^**			
Hyperglycemia	38 (83)	36 (78)	2 (4)
Hypercholesterolemia	27 (59)	27 (59)	0
**Clinical toxicity**			
Asthenia	31 (67)	27 (58)	4 (9)
Nausea	16 (35)	16 (35)	0
Alopecia	14 (30)	14 (30)	0
Mucositis	13 (28)	13 (28)	0
Rash	12 (26)	11 (24)	1 (2)
Pruritis	3 (7)	2 (4)	1 (2)
Constipation	11 (24)	11 (24)	0
Vomiting	9 (20)	9 (20)	0
Dyspnea	9 (20)	8 (17)	1 (2)
Cough	6 (13)	6 (13)	0
Acne	5 (11)	5 (11)	0
Neuropathy	3 (7)	3 (7)	0
Edema	2 (4)	2 (4)	0
Hand–foot syndrome	2 (4)	2 (4)	0

^1^ No febrile neutropenia or bleeding was observed. ^2^ One patient had grade 3 transaminitis.

**Table 3 cancers-14-04509-t003:** Tumor response evaluation (according to RECIST v1.1) after induction chemotherapy among evaluable patients treated at the recommended dose of everolimus of 50 mg/week (N = 41).

Response	N (%)
**Objective response**	31 (75.6)
Complete response	1 (2.4)
Partial response	30 (73.2)
**Stable disease**	9 (22.0)
**Disease progression**	1 (2.4)

## Data Availability

The data presented in this study are available on request from the corresponding author.

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
