# Peer review of "Targeting the Tumor Microenvironment through mTOR Inhibition and Chemotherapy as Induction Therapy for Locally Advanced Head and Neck Squamous Cell Carcinoma: The CAPRA Study"

_cancers, 2022, doi:10.3390/cancers14184509_

Round 1

Reviewer 1 Report

Authors should be congratulated on well designed and properly conducted study in difficult patient population. 

There is a couple of remarks that should be taken into consideration:

- lines 44 and 45: The combination ... achieved major tumor control. This claim is not appropriate as the study focused on response rate and there was no follow-up after induction chemotherapy,

- lines 60 - 64: ... several phase II trials ... showed ... Quotations should be provided,

-lines 310 - 313: ... heavy consumption of tobacco ... did not plan ... human papilloma virus testing... This reasoning does not sound appropriate in view  of standard recommendation of IHC p16 testing in oropharyngeal cancer which composed the half of patients. 

- lines 238 - 240 and 263 - 264: ... major responses in several patients with bulky...lesions ... followed by conclusion: ...Major responses occured  in bulky... This concluding statement would require accurate data like assessment of hypoxia, definition of bulky T lesions and proportion of patients with bulky (N3) neck involvement with major response. 

Author Response

02nd September 2022,

Dear Reviewer 1,

The authors would like to thank you for your time and interest in our work, and for your comments contributing to improve our manuscript. Please find below a point-by-point response to your comments.

For convenience, a red-tracked changes revised version of the manuscript is also provided.

With kind regards,

Sincerely,

Diane EVRARD, MD & Sandrine FAIVRE, MD, PhD

On behalf of the co-authors Clément Dumont, Michel Gatineau, Jean-Pierre Delord, Jérôme Fayette, Chantal Dreyer, Annemilai Tijeras-Raballand, Armand de Gramont, Jean-François Delattre, Muriel Granier, Nasredine Aissat, Marie-Line Garcia-Larnicol, Khemaies Slimane, Benoist Chibaudel, Eric Raymond and Christophe Le Tourneau.

REVIEWER 1 Comments

Answers

lines 44 and 45: The combination ... achieved major tumor control. This claim is not appropriate as the study focused on response rate and there was no follow-up after induction chemotherapy

We thank Reviewer 1 for the comment: rather than long term tumor control, indeed we meant that the induction combination yielded major tumor responses i.e. ³50% by dimensional RECIST criteria; we have corrected the abstract accordingly.

lines 60 - 64: ... several phase II trials ... showed ... Quotations should be provided

As requested by the reviewer, summarized results and references from phase II trials using weekly carboplatin + paclitaxel as induction regimen have been inserted in the introduction of the revised manuscript.

lines 310 - 313: ... heavy consumption of tobacco ... did not plan ... human papilloma virus

testing... This reasoning does not sound appropriate in view of standard recommendation

of IHC p16 testing in oropharyngeal cancer which composed the half of patients

We agree with the reviewer that HPV testing is currently recommended for oropharyngeal cancer. Following the reviewer comment we deleted this sentence in the revised manuscript.

lines 238 - 240 and 263 - 264: ... major responses in several patients with

bulky...lesions ... followed by conclusion: ...Major responses occured in bulky... This

concluding statement would require accurate data like assessment of hypoxia, definition of

bulky T lesions and proportion of patients with bulky (N3) neck involvement with major

response

Thank you for this remark. We did not perform a subgroup analysis focused on bulky or necrotic tumors, but several investigators participating in this study noticed and referred CT scans to PI with major objective responses in several cases of patients with bulky and necrotic tumors as shown in the representative examples in Figure 3 of the manuscript. To avoid any subjective interpretation out of Figure 3, we deleted other specific statements related to bulky or necrotic tumors from the results and the conclusion.

Reviewer 2 Report

This is a very good written paper presenting data from phase I and phase II of the CAPRA study, designed to assess the feasibility of adding everolimus to weekly carboplatin and paclitaxel as induction chemotherapy in patients with untreated locally advanced HNSCC.

They determined the dose of everolimus, the overall response rate (ORR), and the toxicity profile, and assessed tumor and blood biomarkers.

They concluded that the overall response rate of the combined therapy was 75.6% and that they observed a PI3K-AKT-mTOR pathway inhibition and a modulation of the immune tumor microenvironment.

Comments:

Line 133 “may be added to the dose level”: change to past tense.

Line 168: report the number of patients from which biopsies were performed

Figure 1 has some illegible texts, and define DL1 and DL2 in the legend or delete

Line 214: the authors have to explain the reason for the dose reduction in 6 patients.

Figure 3: which were the patients with bulky necrotic and hypoxic lesions, They have to be indicated considering the author commented that the major responses were observed in several of those patients.

Line 261 (figure 4): the data from one patient is not enough to state everolimus specifically targeted the mTOR signaling pathway, they can suggest.

Blood biomarkers section:

Please confirm if no statistical significance was observed compared to the basal value.  If that is correct, why they affirmed that they observed increased plasma levels of IFN-γ, IL-2 and TNF-β, and transient decrease in IL-10 and not of other markers, it must be explained.

What about these markers values in patients that received only  carboplatin-paclitaxel chemotherapy, It can be discussed.

Discussion section:

the authors compared the CAPRA ORR value with ORR TFP treatment, a triplet chemotherapy based on docetaxel, cisplatin, and 5-FU combination (75.6% ORR for CAPRA vs. 60-304 78% for TPF), however, It is necessary to compare with ORR of carboplatin-paclitaxel induction chemotherapy to determine the benefit of including everolimus in the treatment.

The authors mentioned: “CAPRA did not compromise further locoregional approaches since a majority of patients were able to receive platinum-based chemoradiotherapy after induction with CAPRA.” Which patients received platinum-based chemoradiotherapy? Include this information in the text

Finally, the authors mentioned that the assessment of tumor and blood markers aimed at identifying sensitivity/resistance factors to induction treatment, it has to be commented on in the discussion.

Author Response

02nd September 2022,

Dear Reviewer 2,

The authors would like to thank you for your time and interest in our work, and for your comments contributing to improve our manuscript. Please find below a point-by-point response to your comments.

For convenience, a red-tracked changes revised version of the manuscript is also provided.

With kind regards,

Sincerely,

Diane EVRARD, MD & Sandrine FAIVRE, MD, PhD

On behalf of the co-authors Clément Dumont, Michel Gatineau, Jean-Pierre Delord, Jérôme Fayette, Chantal Dreyer, Annemilai Tijeras-Raballand, Armand de Gramont, Jean-François Delattre, Muriel Granier, Nasredine Aissat, Marie-Line Garcia-Larnicol, Khemaies Slimane, Benoist Chibaudel, Eric Raymond and Christophe Le Tourneau.

REVIEWER 2

This is a very good written paper presenting data from phase I and phase II of the CAPRA study, designed to assess the feasibility of adding everolimus to weekly carboplatin and paclitaxel as induction chemotherapy in patients with untreated locally advanced HNSCC.

They determined the dose of everolimus, the overall response rate (ORR), and the toxicity profile, and assessed tumor and blood biomarkers.

They concluded that the overall response rate of the combined therapy was 75.6% and that they observed a PI3K-AKT-mTOR pathway inhibition and a modulation of the immune tumor microenvironment.

REVIEWER 2 Comments

Answers

Line 133 “may be added to the dose level”: change to past tense.

We apologize for this inconsistency; we changed this sentence to past tense accordingly.

Line 168: report the number of patients from which biopsies were performed

Thank you for the remark; we specified the number of patients in which biopsies were performed (9 patients) in the text and the legend of Figure 5.

Figure 1 has some illegible texts, and define DL1 and DL2 in the legend or delete

We apologize for the illegible texts in Fig 1, as suggested we have replaced DL1 and DL2 by the consistent term weekly dose = WD1 and WD 2 and defined it in the legend of Figure 1.

Line 214: the authors have to explain the reason for the dose reduction in 6 patients.

Thank you for this suggestion; we inserted in the text that dose reductions were mainly due to hematological toxicity.

Figure 3: which were the patients with bulky necrotic and hypoxic lesions, They have to be indicated considering the author commented that the major responses were observed in several of those patients.

Thank you for this remark. We did not perform a subgroup analysis focused on bulky or necrotic tumors, but several investigators participating in this study noticed and referred CT scans to PI with major objective responses in several cases of patients with bulky and necrotic tumors as shown in the representative examples in Figure 3 of the manuscript. To avoid any subjective interpretation out of Figure 3, we deleted other specific statements related to bulky or necrotic tumors from the results and the conclusion.

Line 261 (figure 4): the data from one patient is not enough to state everolimus specifically targeted the mTOR signaling pathway, they can suggest.

Thank you for the remark, we replaced “demonstrating” by “suggesting” in the revised manuscript.

Blood biomarkers section

Please confirm if no statistical significance was observed compared to the basal value.  If that is correct, why they affirmed that they observed increased plasma levels of IFN-γ, IL-2 and TNF-β, and transient decrease in IL-10 and not of other markers, it must be explained.

Indeed, given the limited number of patients and the amplitude of the variations, analysis did not reached statistical significance, this has been specified in the result section. We displayed the variation profiles that were observed in this study. Following the suggestion of the reviewer we rephrased the blood biomarkers section using “a trend to increased….” & “a transient trend to decrease in…”

Blood biomarkers section

What about these markers values in patients that received only carboplatin-paclitaxel chemotherapy, It can be discussed.

We agree with the reviewer that it might be interesting to compare our findings with biomarkers variations in patients that received only carboplatin-paclitaxel chemotherapy, but our study was a single arm with no comparator. This limitation was included in the end of discussion section as suggested by the reviewer.

Discussion

the authors compared the CAPRA ORR value with ORR TFP treatment, a triplet chemotherapy based on docetaxel, cisplatin, and 5-FU combination (75.6% ORR for CAPRA vs. 60-78% for TPF), however, It is necessary to compare with ORR of carboplatin-paclitaxel induction chemotherapy to determine the benefit of including everolimus in the treatment.

Our trial was a single arm phase I-II study aiming at defining the safety profile of everolimus-carboplatin-paclitaxel combination, while maintaining antitumor activity in the usual range of objective response rate for induction regimen. For information, summarized results and references from previously published phase II trials using weekly carboplatin + paclitaxel as induction regimen have been inserted in the introduction and discussion of the revised manuscript. The limitation of our single arm trial has also been discussed at the end of the discussion.

Discussion

The authors mentioned: “CAPRA did not compromise further locoregional approaches since a majority of patients were able to receive platinum-based chemoradiotherapy after induction with CAPRA.” Which patients received platinum-based chemoradiotherapy? Include this information in the text

Among 46 patients who received everolimus at 50 mg/week, 41 patients (84%) were able to receive further platinum-based chemoradiotherapy after CAPRA treatment. Following the reviewer’s remark, we included this information in the results (End of paragraph 3.3 “Activity”) and in the discussion.

Discussion

Finally, the authors mentioned that the assessment of tumor and blood markers aimed at identifying sensitivity/resistance factors to induction treatment, it has to be commented on in the discussion.

We agree with the reviewer that data obtained from translational research in this trial allow to report descriptive variations of tumor and blood biomarkers rather than identifying sensitivity/resistance factors to induction treatment. This has been mentioned in the end of the discussion as part of this study limitations.

Reviewer 3 Report

The article by Diane Everard et al. designed a clinical trial and investigated a possible induction treatment that combines Everolimus (an mTOR inhibitor), carboplatin, and paclitaxel chemotherapy. This clinical trial study primarily focused on evaluating the combination
treatment's safety profile and maximum-tolerated dose. The antitumor activities for advanced head and neck cancer patients were also discussed in this study, and the authors concluded promising tumor response results. With the collected blood samples and tissues, the authors also conducted biomarker analysis which validated mTOR signaling inhibition and established the cytokine profiles, indicating an early increase in plasma levels of immunogenic Th1 cytokines, suggesting a facilitating role for antitumor immunity at early time points. Overall, this manuscript is well written and easy to read.
Question:
Everolimus was evaluated in a clinical trial(NCT01111058), which found that Everolimus was not active as monotherapy in unselected patients with recurrent/metastatic HNSCC. Here, in your single-arm clinical study, there is no placebo control. It is difficult to know if the induction
treatment is necessary and has higher efficacy than chemotherapy alone. Do you have ongoing studies that address this concern, or what do you think about this concern?

Author Response

02nd September 2022,

Dear Reviewer 3,

The authors would like to thank you for your time and interest in our work, and for your comments contributing to improve our manuscript. Please find below a point-by-point response to your comments.

For convenience, a red-tracked changes revised version of the manuscript is also provided.

With kind regards,

Sincerely,

Diane EVRARD, MD & Sandrine FAIVRE, MD, PhD

On behalf of the co-authors Clément Dumont, Michel Gatineau, Jean-Pierre Delord, Jérôme Fayette, Chantal Dreyer, Annemilai Tijeras-Raballand, Armand de Gramont, Jean-François Delattre, Muriel Granier, Nasredine Aissat, Marie-Line Garcia-Larnicol, Khemaies Slimane, Benoist Chibaudel, Eric Raymond and Christophe Le Tourneau.

REVIEWER 3

The article by Diane Everard et al. designed a clinical trial and investigated a possible induction treatment that combines Everolimus (an mTOR inhibitor), carboplatin, and paclitaxel chemotherapy. This clinical trial study primarily focused on evaluating the combination treatment's safety profile and maximum-tolerated dose. The antitumor activities for advanced head and neck cancer patients were also discussed in this study, and the authors concluded promising tumor response results. With the collected blood samples and tissues, the authors also conducted biomarker analysis which validated mTOR signaling inhibition and established the cytokine profiles, indicating an early increase in plasma levels of immunogenic Th1 cytokines, suggesting a facilitating role for antitumor immunity at early time points. Overall, this manuscript is well written and easy to read.

REVIEWER 3 Comments

Answers

Everolimus was evaluated in a clinical trial(NCT01111058), which found that Everolimus was not active as monotherapy in unselected patients with recurrent/metastatic HNSCC. Here, in your single-arm clinical study, there is no placebo control. It is difficult to know if the induction

treatment is necessary and has higher efficacy than chemotherapy alone. Do you have ongoing studies that address this concern, or what do you think about this concern?

We agree with the reviewer that our study was a single arm experimental cohort with no comparator arm, making not possible to compare everolimus-based combination versus chemotherapy alone. For the moment, no randomized phase II or III has been launched but such future studies could solve this question. Despite the low activity of everolimus single agent, we designed our trial with the rationale of potential synergistic effects with chemotherapy by targeting the hypoxic microenvironment that could activate PI3K-AKT-mTOR pathway. The main objective of our phase I-II study was to determine the safety profile of everolimus-carboplatin-paclitaxel combination, while maintaining antitumor activity in the usual range of objective response rate for induction regimens. The limitation of no control arm and related issues was included in the end of the discussion section as suggested by the reviewer.

Reviewer 4 Report

The authors show that combination of weekly induction treatment with carboplatin-paclitaxel and mTOR inhibitor everolimus features a good safety profile and a high overall response rate (ORR) (75.6%). Translational data showed decreased p-S6K immunostaining, a surrogate marker of mTORC1 inhibition. Overall, the authors’ methodology and presentation of the study is clear and easy to follow. As a reader, I thought that a few important questions remained unanswered that are important to support the conclusions of the study:

-Line 58: The authors explain that TPF treatment has a high rate of tumor recurrence. In the study only acute ORR is reported, thus we cannot know how the tumor recurrence parameters compare between TPF and CAPRA interventions.

-Line 62: The authors claim that the ORR carboplatin-paclitaxel has been previously reported as 66-89%. This may suggest that everolimus addition does not significantly enhance ORR beyond carboplatin-paclitaxel alone. Importantly, this statement does not include references that I definitely would like to consult in order to assess the differences in efficacy between carboplatin-placitaxel and carboplatin-paclitaxel-everolimus.

-Line 69: Potentially, the action of everolimus might favor a lower rate of tumor recurrence whereas its ability to enhance ORR might be reduced compared to induction chemotherapy alone. I understand this notion might be out of scope for this study but the authors should be encouraged to follow up on recurrence parameters for the experimental cohort as this might give very valuable data.

-Section 3.3: Since this study does not include a control cohort for everolimus that is treated with carboplatin and paclitaxel alone, I need to highlight again that citing studies that have investigated the therapy is crucial to understand the potential added benefit of introducing everolimus.

-Section 3.4: p-S6K is a useful surrogate for mTORC1 activity, however, assessment of changes in AKT and phosphorylation of AKT upon everolimus would be of great interest. In particular, it has been documented, in cancer tissues as well, that inhibiting mTORC1 can lead to lifting of negative feedback of S6K1 on the mTORC2-AKT pathway which ultimately leads to compensatory hyperactivation of AKT (https://doi.org/10.1038/bjc.2016.25, https://doi.org/10.1007/s00428-002-0751-5). These findings raise concerns about the long-term response to mTORC1 inhibition in tumors that rely on AKT hyperactivation.

-Conclusions: Not enough data regarding response to treatment in bulky necrotic tumor lesions in TPF or carboplatin-paclitaxel therapies have been presented to extract conclusions regarding the benefit of everolimus in this particular parameter.

The authors show that combination of weekly induction treatment with carboplatin-paclitaxel and mTOR inhibitor everolimus features a good safety profile and a high overall response rate (ORR) (75.6%). Translational data showed decreased p-S6K immunostaining, a surrogate marker of mTORC1 inhibition. Overall, the authors’ methodology and presentation of the study is clear and easy to follow. As a reader, I thought that a few important questions remained unanswered that are important to support the conclusions of the study:

-Line 58: The authors explain that TPF treatment has a high rate of tumor recurrence. In the study only acute ORR is reported, thus we cannot know how the tumor recurrence parameters compare between TPF and CAPRA interventions.

-Line 62: The authors claim that the ORR carboplatin-paclitaxel has been previously reported as 66-89%. This may suggest that everolimus addition does not significantly enhance ORR beyond carboplatin-paclitaxel alone. Importantly, this statement does not include references that I definitely would like to consult in order to assess the differences in efficacy between carboplatin-placitaxel and carboplatin-paclitaxel-everolimus.

-Line 69: Potentially, the action of everolimus might favor a lower rate of tumor recurrence whereas its ability to enhance ORR might be reduced compared to induction chemotherapy alone. I understand this notion might be out of scope for this study but the authors should be encouraged to follow up on recurrence parameters for the experimental cohort as this might give very valuable data.

-Section 3.3: Since this study does not include a control cohort for everolimus that is treated with carboplatin and paclitaxel alone, I need to highlight again that citing studies that have investigated the therapy is crucial to understand the potential added benefit of introducing everolimus.

-Section 3.4: p-S6K is a useful surrogate for mTORC1 activity, however, assessment of changes in AKT and phosphorylation of AKT upon everolimus would be of great interest. In particular, it has been documented, in cancer tissues as well, that inhibiting mTORC1 can lead to lifting of negative feedback of S6K1 on the mTORC2-AKT pathway which ultimately leads to compensatory hyperactivation of AKT (https://doi.org/10.1038/bjc.2016.25, https://doi.org/10.1007/s00428-002-0751-5). These findings raise concerns about the long-term response to mTORC1 inhibition in tumors that rely on AKT hyperactivation.

-Conclusions: Not enough data regarding response to treatment in bulky necrotic tumor lesions in TPF or carboplatin-paclitaxel therapies have been presented to extract conclusions regarding the benefit of everolimus in this particular parameter.

Author Response

02nd September 2022,

Dear Reviewer 4,

The authors would like to thank you for your time and interest in our work, and for your comments contributing to improve our manuscript. Please find below a point-by-point response to your comments.

For convenience, a red-tracked changes revised version of the manuscript is also provided.

With kind regards,

Sincerely,

Diane EVRARD, MD & Sandrine FAIVRE, MD, PhD

On behalf of the co-authors Clément Dumont, Michel Gatineau, Jean-Pierre Delord, Jérôme Fayette, Chantal Dreyer, Annemilai Tijeras-Raballand, Armand de Gramont, Jean-François Delattre, Muriel Granier, Nasredine Aissat, Marie-Line Garcia-Larnicol, Khemaies Slimane, Benoist Chibaudel, Eric Raymond and Christophe Le Tourneau.

REVIEWER 4

The authors show that combination of weekly induction treatment with carboplatin-paclitaxel and mTOR inhibitor everolimus features a good safety profile and a high overall response rate (ORR) (75.6%). Translational data showed decreased p-S6K immunostaining, a surrogate marker of mTORC1 inhibition. Overall, the authors’ methodology and presentation of the study is clear and easy to follow. As a reader, I thought that a few important questions remained unanswered that are important to support the conclusions of the study:

REVIEWER 4 Comments

Answers

Line 58: The authors explain that TPF treatment has a high rate of tumor recurrence. In the study only acute ORR is reported, thus we cannot know how the tumor recurrence parameters compare between TPF and CAPRA interventions.

We agree with the reviewer that mentioning the rate of tumor recurrence following TPF induction therapy might be confusing with regards to the objectives of CAPRA trial, focusing on safety, ORR, and translational research. Following the reviewer comment we deleted this sentence in the revised manuscript.

Line 62: The authors claim that the ORR carboplatin-paclitaxel has been previously reported as 66-89%. This may suggest that everolimus addition does not significantly enhance ORR beyond carboplatin-paclitaxel alone. Importantly, this statement does not include references that I definitely would like to consult in order to assess the differences in efficacy between carboplatin-paclitaxel and carboplatin-paclitaxel-everolimus.

As requested by the reviewer, summarized results and references from phase II trials using weekly carboplatin + paclitaxel as induction regimen have been inserted in the introduction of the revised manuscript.

Line 69: Potentially, the action of everolimus might favor a lower rate of tumor recurrence whereas its ability to enhance ORR might be reduced compared to induction chemotherapy alone. I understand this notion might be out of scope for this study but the authors should be encouraged to follow up on recurrence parameters for the experimental cohort as this might give very valuable data.

Thank you for this remark, indeed our single arm trial did not intent to compare everolimus-carboplatin-paclitaxel combination to induction chemotherapy alone, but we will keep contact with co-investigators to follow on recurrence parameters in this phase I-II experimental cohort.

Section 3.3: Since this study does not include a control cohort for everolimus that is treated with carboplatin and paclitaxel alone, I need to highlight again that citing studies that have investigated the therapy is crucial to understand the potential added benefit of introducing everolimus.

Our trial was a single arm phase I-II study aiming at defining the safety profile of everolimus-carboplatin-paclitaxel combination, while maintaining antitumor activity in the usual range of objective response rate for induction regimen. For information and as mentioned above (cf comment line 62), summarized results and references from previously published phase II trials using weekly carboplatin + paclitaxel as induction regimen have been inserted in the introduction of the revised manuscript.

Section 3.4: p-S6K is a useful surrogate for mTORC1 activity, however, assessment of changes in AKT and phosphorylation of AKT upon everolimus would be of great interest. In particular, it has been documented, in cancer tissues as well, that inhibiting mTORC1 can lead to lifting of negative feedback of S6K1 on the mTORC2-AKT pathway which ultimately leads to compensatory hyperactivation of AKT (https://doi.org/10.1038/bjc.2016.25, https://doi.org/10.1007/s00428-002-0751-5). These findings raise concerns about the long-term response to mTORC1 inhibition in tumors that rely on AKT hyperactivation.

We agree with the reviewer that assessing phosphorylation of AKT upon everolimus would be of great interest to characterize compensatory hyperactivation of AKT following mTORC1 inhibition. Due to limited material from paired biopsies, we had to select a limited number of biomarkers in this study, but we will keep the suggestion of including p-AKT as an additional biomarker of interest in potential future trials with everolimus to assess the feedback loop through mTORC2-AKT pathway. We have included this remark in the discussion.

Conclusions: Not enough data regarding response to treatment in bulky necrotic tumor lesions in TPF or carboplatin-paclitaxel therapies have been presented to extract conclusions regarding the benefit of everolimus in this particular parameter

Thank you for this remark. We did not perform a subgroup analysis focused on bulky or necrotic tumors, but several investigators participating in this study noticed and referred CT scans to PI with major objective responses in several cases of patients with bulky and necrotic tumors as shown in the representative examples in Figure 3 of the manuscript. To avoid any subjective interpretation out of Figure 3, we deleted other specific statements related to bulky or necrotic tumors from the results and the conclusion.

Round 2

Reviewer 3 Report

Thank you for the detailed response. I have no further questions

Author Response

Dr Diane EVRARD

                                                                                             Department of Head and Neck Surgery

                                                Bichat University Hospital, Assistance Publique—Hôpitaux de Paris

                                                                                    46 Rue Henri Huchard, 75877 Paris, France,

                                                                                                                 Telephone: +33677846800

                                                                                                         email: [email protected]

12nd September 2022,

Dear Reviewer,

The authors would like to thank you for your time and interest in our work, and for your comments contributing to improve our manuscript.

With kind regards,

Sincerely,

Diane EVRARD, MD & Sandrine FAIVRE, MD, PhD

On behalf of the co-authors Clément Dumont, Michel Gatineau, Jean-Pierre Delord, Jérôme Fayette, Chantal Dreyer, Annemilai Tijeras-Raballand, Armand de Gramont, Jean-François Delattre, Muriel Granier, Nasredine Aissat, Marie-Line Garcia-Larnicol, Khemaies Slimane, Benoist Chibaudel, Eric Raymond and Christophe Le Tourneau.

Reviewer 4 Report

The authors' single arm phase I-II study successfully characterizes the safety profile of everolimus-carboplatin-paclitaxel and demonstrates that the treatment combination maintains antitumor activity in a range similar to TPF and carboplatin-paclitaxel. The authors are clear and accurate in reporting their findings.

Arising from the data shown here, there is no clear, established advantage of everolimus-carboplatin-paclitaxel treatment over carboplatin-paclitaxel or TPF treatments ORR. A statement in the discussion that express future directions aiming to investigate the advantage of everolimus supplementation (lower recurrence rate? better response in a subset of hyperactive AKT-dependent tumors? etc.) would enhance the manuscript.

Author Response

Dr Diane EVRARD

                                                                                             Department of Head and Neck Surgery

                                                Bichat University Hospital, Assistance Publique—Hôpitaux de Paris

                                                                                    46 Rue Henri Huchard, 75877 Paris, France,

                                                                                                                 Telephone: +33677846800

                                                                                                         email: [email protected]

September 12, 2022

Dear Reviewer,

We thank you giving us the opportunity to resubmit with minor revisions our manuscript “Targeting the tumor microenvironment through mTOR inhibition and chemotherapy as induction therapy for locally advanced head and neck squamous cell carcinoma: the CAPRA study”.

Please find a second revision integrating changes requested by you and the editor. Based on the comment from Academic Editor, we have provided a separate limitations paragraph at the end of the discussion and added the limitations including your concerns to the conclusions. Corrections have been tracked in red in a revised version of the original manuscript.

We hope that this revised manuscript has improved and could be now acceptable for publication in Cancers.

With kind regards,

Sincerely,

Diane EVRARD, MD & Sandrine FAIVRE, MD, PhD

On behalf of the co-authors Clément Dumont, Michel Gatineau, Jean-Pierre Delord, Jérôme Fayette, Chantal Dreyer, Annemilai Tijeras-Raballand, Armand de Gramont, Jean-François Delattre, Muriel Granier, Nasredine Aissat, Marie-Line Garcia-Larnicol, Khemaies Slimane, Benoist Chibaudel, Eric Raymond and Christophe Le Tourneau.